# Transcriptomic and Biochemical Analysis of the Antimicrobial Mechanism of Lipopeptide Iturin W against *Staphylococcus aureus*

**DOI:** 10.3390/ijms25189949

**Published:** 2024-09-15

**Authors:** Yingyu Ji, Chaomin Sun, Shimei Wu

**Affiliations:** 1College of Life Sciences, Qingdao University, Qingdao 266071, China; wojiyingyu@163.com; 2CAS Key Laboratory of Experimental Marine Biology, Institute of Oceanology, Chinese Academy of Sciences, Qingdao 266071, China; sunchaomin@qdio.ac.cn; 3Laboratory for Marine Biology and Biotechnology, Qingdao National Laboratory for Marine Science and Technology, Qingdao 266071, China; 4Center of Ocean Mega-Science, Chinese Academy of Sciences, Qingdao 266071, China

**Keywords:** *Staphylococcus aureus*, lipopeptide, antimicrobial, virulence

## Abstract

*Staphylococcus aureus* is one of the most serious pathogens threatening food safety and public health. We have previously showed that iturin W exhibited obvious antifungal activity on plant pathogens. In the present study, we found iturin W, especially C_14_ iturin W, showed strong antimicrobial activity against *S. aureus*, and the antimicrobial mechanism of C_14_ iturin W was further investigated by transcriptomic analysis and a related biochemical experiment. The results showed that C_14_ iturin W can reduce the expression levels of genes associated with the reactive oxygen species (ROS) scavenging enzyme and genes involved in arginine biosynthesis, thus leading to the increase in ROS levels of *S. aureus*. Furthermore, C_14_ iturin W can also interfere with proton dynamics, which is crucial for cells to regulate various biological possesses. Therefore, ROS accumulation and change in proton motive force are import ways for C_14_ iturin W to exert the antimicrobial activity. In addition, C_14_ iturin W can also reduce the expression levels of genes related to virulence factors and decrease the production of enterotoxins and hemolysins in *S. aureus*, indicating that C_14_ iturin W has a good potential in food and pharmaceutical fields to reduce the harm caused by *S. aureus* in the future.

## 1. Introduction

Foodborne pathogens, such as norovirus, *Salmonella*, *Shigella*, *Campylobacter*, *Clostridium perfringens*, *Listeria monocytogenes* and *Staphylococcus aureus*, are a major threat to food safety, which can cause serious human health problems and even lead to death [1,2]. *S. aureus* is a well-known pathogen capable of producing toxins during bacterial growth in contaminated food, and if people digest the corresponding food, it often leads to food poising [3,4]. It was estimated that *S. aureus* caused about 20–25% of foodborne bacterial outbreaks in China [5]. *S. aureus* causes foodborne disease mainly by producing different toxins, such as staphylococcal enterotoxins (SEs), hemolysins and leukocidins. SEs are the major cause of staphylococcal food poisoning, which can lead to significant gastrointestinal symptoms such as nausea, vomiting, abdominal pain, cramps, diarrhea and even death [6]. Additionally, hemolysins can attack cell membranes by forming pores, resulting in cell death and causing intestinal inflammation [7]. Various virulence determinants, such as enterotoxins, hemolysins and leukocidins, were reported in milk samples collected from mastitic household dairy cows, and some *S. aureus* strains harboring hemolysin genes were also isolated from the milk of healthy goats [8,9]. Furthermore, unreasonable use of antibiotics for antimicrobial treatment or growth promotion has led to the development of resistance in *S. aureus*, such as methicillin-resistant *S. aureus* (MRSA), which has been detected in contaminated food recently [10]. Therefore, new antimicrobial agents against *S. aureus*, especially against MRSA, are urgently needed.

With the wide use of antibiotics and emergence of drug-resistant pathogen, antimicrobial peptides (AMPs) are regarded as a new generation of antibiotics to combat resistant pathogens [11]. AMPs are naturally occurring small proteins or peptides that exhibit strong antimicrobial activity against various bacteria, fungi and viruses. Unlike traditional antibiotics with only one target, AMPs can use different mechanisms to inhibit bacteria, such as disturbing the cell membrane structure and targeting intracellular macromolecules after entering the cell membrane [12]. Compared with traditional antibiotics, AMPs have the advantages of low toxicity, strong thermal stability, low molecular weight and the lack of resistance [13]. *Bacillus* spp. is a Gram-positive bacterium that can produce various bioactive compounds, including AMPs [14]. There are two types of antimicrobial peptides derived from *Bacillus*, including ribosomal synthesized peptides (such as bacteriocins) and non-ribosomal synthesized peptides (such as lipopeptides and polyketide peptides) [15]. Lipopeptides are a type of antimicrobial peptide with amphiphilic structures, including hydrophobic cyclic peptides and hydrophilic fatty acid chains. They have various biological activities such as antifungal, antibacterial, antiviral, anti-tumor, anti-inflammatory, etc. Therefore, its development and application in fields such as agriculture, medicine and food industry are of great significance [16].

In order to screen potential strains that can significantly inhibit the cell growth of *S. aureus*, the antimicrobial activities of a series of marine strains were detected in this study, and *Bacillus* sp. wsm-1 was found to have the strongest inhibitory activity against *S. aureus*. In our previous study, *Bacillus* sp. wsm-1 was identified and proved to produce lipopeptide iturin W, which consists of two isoforms C_14_ iturin W and C_15_ iturin W, and both of them can effectively inhibit a variety of plant pathogenic fungi [17]. In this study, we found that C_14_ iturin W could obviously inhibit the growth of *S. aureus*. The corresponding mechanism was investigated through electron microscopy observation and transcriptomic analysis, and related biochemical experimental validation was also carried out in our present study.

## 2. Results and Discussion

### 2.1. Screening, Purification and Identification of the Antimicrobial Agents

In order to obtain strains that have a strong inhibitory effect on MRSA CCARM 3090, 200 marine bacterial strains were isolated and screened against it, and *Bacillus* sp. wsm-1 was found to exhibit the strongest antimicrobial activity against *S. aureus*, and this strain was demonstrated to have strong anti-fungal activity by producing lipopeptide iturin W in our previous study [17]. The antimicrobial agents against *S. aureus* from *Bacillus* sp. wsm-1 was further purified, and two active fractions with retention time about 20.420 min and 25.725 min were detected at the RP-HPLC step (Figure 1A), which were approximately the same as those of C_14_ iturin W and C_15_ iturin W that we previously isolated under the same elution conditions, so the purified antimicrobial agents were presumed as C_14_ iturin W and C_15_ iturin W, respectively.

To further confirm that the purified antimicrobial agents were C_14_ iturin W and C_15_ iturin W, the products eluted at 20.420 min and 25.725 min were analyzed by MS/MS analysis. For the fraction eluted at 20.420 min, the maximum fragment of it is 1043.55, which is identical to the single protonated ion [M + H]+ of C_14_ iturin W in our previous report. Starting from the N terminus, fragments of b ion in descending order were 1043.55 (b8), 915.49 (b7), 801.45 (b6), 638.39 (b5), 541.33 (b4), 427.29 (b3), 340.26 (b2) and 226.22 (b1), and the differences between the values were exactly the mass of ion fragments of Gln, Asn, Tyr, Pro, Asn, Ser and Asn (Figure 1B), so the sequence of it is the β-amino fatty acid-Asn-Ser-Asn-Pro-Tyr-Asn-Gln, and the amino fatty acid contains 14 carbon groups. For the fraction eluted at 25.725 min, the maximum fragment of it is 1057.57, which is identical to the single protonated ion [M + H]+ of C_15_ iturin W in our previous report. Starting from the N terminus, fragments of b ion in descending order were 1057.57 (b8), 929.51 (b7), 815.47 (b6), 652.40 (b5), 555.35 (b4), 441.31 (b3), 354.28 (b2) and 240.23 (b1), and the differences between the values were also exactly the mass of ion fragments of Gln, Asn, Tyr, Pro, Asn, Ser and Asn (Figure 1C), so the sequence of it is also the β-amino fatty acid-Asn-Ser-Asn-Pro-Tyr-Asn-Gln, but the amino fatty acid contains 15 carbon groups. Therefore, the sequences of the two antimicrobial agents were identical, and they have only one methylene group different, which are identical to those of C_14_ iturin W and C_15_ iturin W as we previously studied; so the purified antimicrobial agents were presumed as C_14_ iturin W and C_15_ iturin W, respectively.

### 2.2. Antimicrobial Activity Assay of C_14_ Iturin W and C_15_ Iturin W

In order to detect the antimicrobial activities of C_14_ iturin W and C_15_ iturin W, their inhibition rate against MRSA CCARM 3090 was measured. As shown in Figure 2A,B, under the same concentration, the growth inhibitory rate of C_14_ iturin W was much higher than that of C_15_ iturin W. Especially at the concentration of 32 μg/mL, the inhibition rate of C_14_ iturin W exceeded 90%, while the inhibition rate of C_15_ iturin W was about only 50%. Since the antimicrobial activity of C_14_ iturin W was higher than that of C_15_ iturin W, C_14_ iturin W was used to investigate further in the following experiment. To further elucidate the antibacterial activity of C_14_ iturin W against *S. aureus*, the minimum inhibitory concentration (MIC) of C_14_ iturin W against four different types of *S. aureus* was detected. As shown in Table 1, the MICs of C_14_ iturin W against MRSA CCARM 3090 and MRSA QD-1 are 64 μg/mL, while MICs of C_14_ iturin W against *S. aureus* ATCC 25923 and *S. aureus* QD-2 are 32 μg/mL.

It has been reported that fatty acid chains are important for their biological activity [18]. In our study, the antimicrobial activity of C_14_ iturin W is higher than that of C_15_ iturin W, which is consistent with previous reports that lipopeptide isoforms with short fatty acid chains are more selective and potent towards bacteria [19].

### 2.3. Morphological and Ultrastructural Changes of MRSA CCARM 3090 Caused by C_14_ Iturin W

To investigate the effect of C_14_ iturin W on MRSA CCARM 3090, the changes in the morphology and ultrastructure of MRSA CCARM 3090 was observed by scanning electron microscopy (SEM) and transmission electron microscopy (TEM) after treated with C_14_ iturin W. The results showed that the cell surface of the control group was smooth, and the cell distribution was scattered and not agglomerated (Figure 3A), but the cells were closely clustered after treated with C_14_ iturin W, the cell surface was no longer smooth and some holes were observed on the surface (Figure 3B). TEM observation showed that the cell wall thickness of untreated *S. aureus* was uniform and there was a tight junction between the cell wall and the cell membrane (Figure 3C), while the cell wall became swollen after treated with C_14_ iturin W, gaps between the cell wall and cell membrane were observed and the cell content became sparse in some cells (Figure 3D).

Staphylococcus cell walls are not only responsible for bacterial structural integrity but also closely related to the interaction with the host’s innate immune system [20], and the cell membrane protects *S. aureus* from external pressure and antimicrobial agents [21]. Our results by electron microscopy observation indicated that lipopeptide C_14_ iturin W might kill the *S. aureus* cells by changing the cell wall and cell membrane, which is consistent with previous reports about lipopeptide bacillomycin D against the plant pathogen *Fusarium graminearum* [22].

### 2.4. Transcriptomic Profiling of MRSA CCARM 3090 after Treated with C_14_ Iturin W

In order to investigate the inhibitory mechanism of C_14_ iturin W, transcriptomic analysis of MRSA CCARM 3090 was performed after treated with C_14_ iturin W. Compared with the control group without C_14_ iturin W treatment, 57 differentially expressed genes (DEGs) were up-regulated, and 88 DEGs were down-regulated in the group treated with the 4 µg/mL of C_14_ iturin W (Figure 4A), and 225 DEGs were up-regulated, and 273 genes were down-regulated in the group treated with 24 µg/mL of C_14_ iturin W (Figure 4B). There were 27 DEGs with expression levels jointly up-regulated (Figure 4C) and 62 DEGs with expression levels jointly down-regulated (Figure 4D) after *S. aureus* was treated with 4 µg/mL and 24 µg/mL of C_14_ iturin W, respectively.

The GO enrichment and KEGG enrichment analyses were also performed after MRSA CCARM 3090 cells were treated with 24 µg/mL of C_14_ iturin W. As shown in Figure 5A, the DEGs classified as a specific cellular component were mainly associated with an integral component of membrane, a membrane part and a membrane, and the expression levels of the relevant parts of DEGs were down-regulated in MRSA CCARM 3090 cells, which can be mutually corroborated with the results observed by electron microscope. In addition, DEGs with down-regulated expression levels on biological processes are involved in a number of amino acid anabolic pathways, which are detrimental to the growth and survival of MRSA CCARM 3090. The KEGG pathway analysis showed that genes involved in amino acid biosynthesis, infection and arginine biosynthesis were significantly down-regulated (Figure 5B), which indicate that C_14_ iturin W not only can inhibit the growth and metabolism of MRSA CCARM 3090 but also can reduce the infection of MRSA CCARM 3090. Through the analysis of transcriptome data, we can speculate the pathways and genes significantly changed by C_14_ iturin W so as to have a preliminary understanding of the mechanism of C_14_ iturin W against *S. aureus*.

### 2.5. Functional Analysis of DEGs

Based on GO and KEGG enrichment analyses, detailed functional analysis of significantly changed DEGs was performed. As shown in Table 2, some genes related to ROS accumulation and virulence factors were significantly changed.

#### 2.5.1. DEGs Involved in ROS Accumulation and Proton Motive Force

It is reported that high concentrations of ROS can induce oxidative stress, leading to DNA damage, lipid peroxidation and protein oxidation, thus leading to cell death [23], and dramatic down-regulation of gene expression on the arginine synthesis pathway leads to arginine deficiency and ROS accumulation [24]. Furthermore, arginine to ornithine conversion was also coupled to the proton motive force [25,26]. In our study, the gene *ILP78_10355* encoding arginine succinate synthetase and the gene *argH* encoding arginine succinate lyase were significantly down-regulated, up to 2^7^-fold, in C_14_ iturin W-treated cells, and the gene *argF* encoding ornithine carbamoyl-transferase was also obviously down-regulated (Table 2). In addition, some genes encoding ROS clearance enzymes, such as the gene *ILP78_00430* encoding superoxide dismutase (SOD) and the gene *ILP78_01305* encoding glutathione peroxidase (GSH-Px), were significantly down-regulated in C_14_ iturin W-treated cells, which are reported to be responsible for converting the formed peroxides into less toxic or harmless substances by redox [27,28].

The *mnh* manipulator (*mnhABCDEFG*) of *S. aureus* encodes a Na^+^/H^+^ antiporter [29], and Mnh anti-transporter proteins play an important role in the maintenance of prokaryotic cytoplasmic pH, Mnh2 being able to enable the exchange of Na^+^/H^+^ and K^+^/H^+^ cations [30]. In our study, the expression levels of Mnh-related genes *mnh2A* and *mnh2B* were also down-regulated in C_14_ iturin W-treated MRSA CCARM 3090 cells (Table 2), which might lead to a diminished ion exchange, inability of the ionic concentration inside and outside of the cell membrane to reach resting potential and alteration in the membrane proton motive force. In addition, the down-regulated Mnh-related genes lead to changes in osmotic pressure, which may be related to the previously reported formation of ROSs [31].

The ROS is produced in cells during metabolic processes, but interference with intracellular oxidoreductases and a direct increase of ROSs can cause ROS/GSH imbalance, leading to harmful oxidation and chemical modification of biomolecules, ultimately resulting in a cell cycle arrest and proliferation inhibition, and even inducing cell death [32]. After treatment with C_14_ iturin W, the expression levels of ROS scavenging enzymes were down-regulated, leading to the inability to clear ROSs in a timely manner and inducing cell death. In addition, bacterial proton motive force is crucial for cells to regulate various biological possesses, such as adenosine triphosphate synthesis and active transport of molecules [33], and the proton motive force was also changed by C_14_ iturin W. Therefore, C_14_ iturin W might exert its antimicrobial activity by inducing ROS accumulation and disturbing the proton motive force of the cell membrane.

#### 2.5.2. DEGs Involved in Virulence of MRSA CCARM 3090

Since hemolysins are an important virulence factor of *S. aureus* [34], the expression levels of genes related to hemolysins in *S. aureus* were analyzed. As shown in Table 2, the expression levels of the genes *hlgA*, *hlgB* and *hlgC,* encoding the γ-hemolysin, and the gene *hyl*, encoding the α-hemolysin, were decreased obviously, especially after treated with 24 μg/mL of C_14_ iturin W, the expression levels of related genes were down-regulated up to 2^5^-fold. And the expression level of the gene *spa*, which encodes the staphylococcal protein A, was also significantly down-regulated in C_14_ iturin W-treated cells, and the staphylococcal protein A is the main surface protein and an important virulence factor of *S. aureus* [35].

The ability of *S. aureus* to cause persistent infection is related to its ability to evade or inactivate the host immune response. One of the immune escape strategies is to secrete proteins that inhibit complement activation, such as the extracellular complement binding protein (Ecb) and extracellular fibrinogen binding protein (Efb) [36,37]. In the present study, the expression levels of *ecb* and *efb* genes in cells were significantly down-regulated after C_14_ iturin W treatment. In addition, the expression levels of the genes *lukG* and *lukH* encoding leukocidin, which made human phagocytes lyse and was thought to contribute to immune escape in *S. aureus* [38], were also significantly down-regulated in C_14_ iturin W-treated MRSA CCARM 3090. In addition, it was previously reported that the cell wall-anchoring enzyme adenosine synthase A (AdsA) is an important immune evasion factor of *S. aureus* and is required for staphylococcal survival in the bloodstream [39], and the expression level of the gene *adsA* was also decreased in *S. aureus* cells after treated with C_14_ iturin W.

Therefore, C_14_ iturin W not only can attenuate the virulence of *S. aureus*, such as the hemolysin and staphylococcal protein A, but also can reduce its ability to evade the host immune response, thereby reducing the damage of *S. aureus* to host cells and allowing the host’s immune system to also exert its killing effect.

### 2.6. qRT-PCR Verification of the Related DGEs

In order to further confirm our transcriptomic analysis, qRT-PCR was used to detect the effects of C_14_ iturin W on related DEGs involved in ROS accumulation and virulence factors. As shown in Figure 6, the ROS accumulation related genes *argF* and *ILP78_01305*, and the virulence factor related genes *hlgA*, *efb* and *lukG* were all obviously down-regulated after treated with 4 μg/mL and 24 μg/mL of C_14_ iturin W, and the results were consistent with our transcriptomic data generated by RNA-seq, thus confirming the reliability of our transcriptomic data.

### 2.7. Biochemical Functional Validation of DEGs

Since the biosynthesis of arginine was significantly down-regulated in C_14_ iturin W-treated cells, in order to verify the correlation between arginine and C_14_ iturin W, the growth of C_14_ iturin W-treated *S. aureus* was detected in the presence or absence of arginine. As shown in Figure 7A, the growth of *S. aureus* was significantly inhibited by C_14_ iturin W, but obviously rescued by the supplementation of arginine, while the growth of *S. aureus* was similar to the C_14_ iturin W-untreated group regardless of whether arginine was supplemented or not. Therefore, the result further demonstrated that interfering with biosynthesis of arginine was an important way for C_14_ iturin W to exhibit its growth inhibitory activity, which is consistent with our transcriptomic analysis.

In order to investigate whether the number of ROSs was increased after *S. aureus* was treated with C_14_ iturin W, the ROS was detected with fluorescence probe DCFH-DA. As shown in Figure 7B, the ROS levels were obviously increased in a dose-dependent style after *S. aureus* was treated with different concentrations of C_14_ iturin W. However, the ROS levels in *S. aureus* cells were significantly decreased when supplemented with 5 mg/mL of arginine, which indicated that the arginine supplementation can reduce the production of ROSs. Our results are consistent with a previous study that reports that the inhibition of biosynthesis of arginine can lead to ROS production [24].

Based on the transcriptomic analysis, the genes related to Na^+^/H^+^ antiporters were also down-regulated, which might lead to the changes in the cell membrane potential. In order to verify our speculation, the membrane potential of *S. aureus* was detected after treated with C_14_ iturin W. As shown in Figure 7C, the membrane potential of *S. aureus* obviously changed when treated with C_14_ iturin W. In addition, the membrane permeability was also increased in a dose-dependent manner when treated with C_14_ iturin W (Figure 7D). Therefore, we believe that C_14_ iturin W caused damage to the membranes of *S. aureus*, thus causing its dysfunction and cell death.

### 2.8. Inhibition of C_14_ Iturin W on Virulence of MRSA CCARM 3090

Because the hemolysin is an important virulence factor of *S. aureus*, and genes related to the virulence factors of *S. aureus* were significantly down-regulated by C_14_ iturin W, the effect of C_14_ iturin W on the hemolytic activity of *S. aureus* was determined by erythrocyte lysis assay. Firstly, the hemolytic activity of different concentrations of C_14_ iturin W was detected. As shown in the upper line of the microtiter plate of Figure 7E, C_14_ iturin W did not have any hemolytic activity even at the concentration of 64 µg/mL. Then, the hemolytic activity of *S. aureus* was also detected after treated with different concentration of C_14_ iturin W. As shown in the lower line of the microtiter plate of Figure 7E, the supernatant of the *S. aureus* cell exhibited obvious hemolytic activity because the red blood cells dissolved as the positive control and 8 µg/mL of C_14_ iturin W could partially reduce the hemolytic activity of *S. aureus*, and the hemolytic activity of *S. aureus* was significantly inhibited when the concentration of C_14_ iturin W was 16 µg/mL. Therefore, C_14_ iturin W has no hemolytic activity itself and can significantly reduce the virulence of *S. aureus*, indicating that it has a good potential as food preservative in the future.

Since SEs are major virulence factors of staphylococcal food poisoning, the effect of different concentrations of C_14_ iturin W on the production of SEs was detected. As shown in Figure 7F, the concentration of SEs in the control group untreated with C_14_ iturin W was 49.97 pg/mL and reduced to 31.04 and 29.10 pg/mL when the groups were treated with 4 and 8 µg/mL of C_14_ iturin W, respectively. And when the group was treated with 16 µg/mL of C_14_ iturin W, the concentration of SEs was significantly lower than that of the control group, which was only 13.29 pg/mL. Therefore, C_14_ iturin W has the ability to reduce the production of SEs, thereby attenuating the toxicity of *S. aureus*.

*S. aureus* is a pathogenic bacterium that can exist in food and cause food contamination, leading to health problems in animals and humans. The exotoxins it produces, including enterotoxins and hemolysins, can cause mastitis in milk producing ruminants, food poisoning in humans and blood infections [40]. The occurrence of *S. aureus* infection largely depends on various extracellular virulence factors secreted by *S. aureus* [41]. At the concentrations below MIC, C_14_ iturin W also has an obvious inhibitory effect on the production of exotoxins, which indicates that C_14_ iturin W not only can inhibit the growth of *S. aureus* but also can attenuate its virulence.

## 3. Materials and Methods

### 3.1. Screening of Strains with Inhibitory Activity against S. aureus

In order to screen bacteria that can inhibit *S. aureus*, the inhibitory effect of 200 marine strains isolated and purified in our laboratory were detected according to the previous method. The bacteria to be tested were inoculated in a Luria–Bertani (LB) or 2216E medium at 28 °C in a shaker at the speed of 150 rpm, and the fermentation supernatant was obtained by centrifugation at 9000 rpm after incubated for 48 h. The overnight cultures of the indicator strain MRSA CCARM 3090 were inoculated in an LB agar medium and poured into plates, then 100 μL of the cell-free supernatant of the isolated strains was added into the pre-punctured wells in an LB plate. The corresponding LB plate was further incubated at 37 °C for 12–15 h, and a strain with a clear inhibitory zone was considered to have inhibitory activity against *S. aureus*. The indicator strain *S. aureus* was routinely grown in an LB medium and incubated at 37 °C, and the marine bacterial strains were cultured in a 2216E medium or an LB medium and incubated at 28 °C. *S. aureus* used in the experiment included MRSA CCARM 3090, MRSA QD-1, *S. aureus* QD-2 and *S. aureus* ATCC 25923.

### 3.2. Purification of the Antimicrobial Agents S. aureus

The purification of iturin W against *S. aureus* from *Bacillus* sp. wsm-1 was carried out according to the previous method [17]. In brief, the overnight culture of *Bacillus* sp. wsm-1 was inoculated into a nutrient broth (NB) medium and incubated at 28 °C for 48 h at 150 rpm. The samples were centrifuged at 4 °C at 9000 rpm for 20 min, and the cell-free supernatant was collected through centrifugation, then subsequently acidified to pH 2.5 with 6 M HCl and stored overnight at 4 °C. The corresponding precipitate was collected by centrifugation, and the anti- MRSA CCARM 3090 component was extracted with methanol. The crude extracts were further purified by reversed-phase high-performance liquid chromatography (RP-HPLC; Agilent 1260, Santa Clara, CA, USA) with an Eclipse XDB-C18 column (5 µm; 4.6 by 250 mm; Agilent, USA) at 2 mL/min with the following condition: From 0 to 5 min, from the 0% mobile phase B to 70% mobile phase B, then eluted with the 70% mobile phase B for 35 min. Among them, the mobile phase A is composed of water and methanol (90:10, volume ratio), and the mobile phase B is 100% methanol.

### 3.3. Identification of the Purified Antimicrobial Agents

To identify the purified antimicrobial agents, the purified antimicrobial agents were analyzed and identified by tandem mass spectrometry (MS/MS), which was performed using the high-energy-collision-induced dissociation (HCD) mode of a linear ion trap Orbitrap spectrometer (LTQ Orbitrap XL; Thermo Fisher, Waltham, MA, USA) as we described previously [42]. Mass spectrum detection conditions: Negative ion scan with voltage 3 kV, temperature 275 °C, pressure 0.05 mPa, HCD collision energy 30–40 eV and scan range *m*/*z* 100~1100. Results were analyzed with Xcalibur 3.0 (Thermo Fisher).

### 3.4. Antimicrobial Activity Assay

To detect the antimicrobial activity, the active fractions from RP-HPLC were collected, which have more than 95% purity, and the purified agents were dissolved in methanol in 2.56 mg/mL as the stock solution for further experiment. The MICs of the purified agents against *S. aureus* were measured according to previous reports with minor modifications [43]. Briefly, the overnight culture of MRSA CCARM 3090 was diluted to 10^4^–10^5^ CFU/mL, then 190 μL of diluted culture was added to each well of a 96-well microtiter plate, and 10 μL of 2-fold gradient dilutions of C_14_ iturin W was added. The 96-well plates were incubated at 37 °C with agitation at 150 rpm overnight, and the growth was measured by the detection of OD_600_. The group without the addition of C_14_ iturin W was set as the control group. The inhibition rate (%) = (1 − (A/A_0_)) × 100%. A is the OD_600_ value of the group treated with C_14_ iturin W, and A_0_ is the OD_600_ value of the control group.

### 3.5. SEM and TEM Observation

The morphological and ultrastructural changes of *S. aureus* caused by C_14_ iturin W were observed using an SEM and a TEM. The culture of *S. aureus* was diluted with a fresh LB medium to OD_600_ of 0.05, then 24 μg/mL of C_14_ iturin W was added and incubated overnight at 37 °C. Then the cells were collected by centrifugation at 3500 rpm for 10 min. After centrifugation, the supernatant of *S. aureus* was discarded, and the cells were washed three times with PBS buffer, then fixed with 2.5% glutaraldehyde for 8 h and dehydrated with ethanol gradient. Scanning samples were observed by a Hitachi S-3400N scanning electron microscope (Hitachi, Tokyo, Japan). Transmission samples were embedded in plastic resin sections, observed by a transmission electron microscope (HT7700; Hitachi, Japan) with a JEOL JEM 12000 EX (equipped with a field emission gun) at 120 kV. The sample without C_14_ iturin W treatment was used as a control.

### 3.6. Transcriptomic Analysis

To investigate the inhibitory mechanism of the C14 iturin W, the culture of *S. aureus* was diluted with a fresh LB medium to OD_600_ of 0.05, then different concentrations of C_14_ iturin W (0 μg/mL, 4 μg/mL and 24 μg/mL) were added and incubated at 37 °C overnight. Then the cells were centrifuged at 3500 rpm for 10 min, the supernatant was discarded and the cells were washed three times with PBS. Three parallel experimental groups for each sample were set and combined into one sample to be sent to Novogene (Tianjin, China) to perform transcriptomic analysis. The cells were collected, liquid nitrogen was added to grind the cell walls and the total RNAs were extracted using an RNApure Bacteria Kit (DNase I) (CWBio, Beijing, China). In brief, 3 mg of RNA per sample was required for the input material for RNA sample preparation. According to the manufacturer’s recommendations, an NEBNext^®^ Ultra^TM^ Directional RNA Library Prep Kit for Illumina^®^ (NEB, Ipswich, MA, USA) was used to generate sequencing libraries. Differential expression analysis of RNA between two different groups was performed using the DESeq 2 software package (1.20.0) and edgeR v3.24.3, and the negative binomial distribution model was used for a statistical test. The Benjamini and Hochberg’s method was used to make adjustments to the *p*-values. The thresholds for significant differential expression were a corrected *p*-value of 0.005 and a log_2_ (fold change) of 1. GO functional enrichment analysis and KEGG pathway enrichment analysis were carried out by using clusterProfiler software (3.8.1).

### 3.7. Quantitative Real-Time Reverse Transcription PCR (qRT-PCR) Analysis

In order to validate the relative change of gene expression based on transcriptomic analysis, the qRT-PCR analysis was performed. The culture of *S. aureus* was diluted to OD_600_ of 0.05 and then treated with different concentrations of C_14_ iturin W (0 μg/mL, 4 μg/mL and 24 μg/mL) at 37 °C. After incubated overnight, the cells were collected and frozen in liquid nitrogen. Total RNAs were extracted using the RNApure Bacteria Kit (DNase I) (CWBio, Beijing, China) after the cells were grinded in liquid nitrogen, then reverse transcribed into cDNA using All-In-One 5X RT MasterMix (Abm, Richmond, BC, Canada) according to the manufacturer’s instructions. The qRT-PCR was performed using an SYBR green Realtime PCR Master Mix (TOYOBO, Osaka, Japan) and a QuantStudio^TM^ 6 Flex Real-Time PCR System (Thermo Fisher Scientific, USA). Primer design was carried out with Primer-BLAST (http://www.ncbi.nlm.nih.gov/tools/primer-blast/ (accessed on 7 September 2024)), and corresponding primers are listed in Table 3. The 16S rRNA gene of *S. aureus* was used as the internal control. The relative gene expression was calculated using the 2^−ΔΔCt^ method, and each sample contained three replicates.

### 3.8. Detection of the Correlation between Arginine and the C_14_ Iturin W

In order to detect the effect of arginine on the activity of C_14_ iturin W, the overnight culture of *S. aureus* was diluted to 10^4^–10^5^ CFU/mL and treated with 5 mg/mL of arginine, 32 μg/mL of C_14_ iturin W, 5 mg/mL arginine and 32 μg/mL of C_14_ iturin W, respectively. After incubation at 37 °C overnight, and the OD value was measured at 600 nm using an Infinite M1000 Pro Microplate reader (TECAN, Männedorf, Switzerland). Each group was replicated in triple, and the group untreated with C_14_ iturin W was used as a negative control.

### 3.9. Determination of ROS

The cellular ROS detection was performed according to previously described methods with minor modifications [44]. The overnight culture of *S. aureus* cells supplemented with or without 5 mg/mL of arginine was diluted to OD_600_ of 0.3 with PBS, and 2′,7′-Dichlorodihydrofluorescein diacetate (DCFH-DA, Solarbio, Beijing, China) was added at a final concentration of 10 μM, then incubated at 37 °C in a dark environment for 1 h. After that, the cells were washed three times with PBS, then they were treated with different concentrations of C_14_ iturin W and further incubated for 5 min. The fluorescence results were obtained by taking readings at 525 nm emission and 488 nm excitation on the Infinite M1000 Pro Microplate reader (TECAN, Switzerland). Each group was replicated in triple, and the group untreated with C_14_ iturin W was used as a negative control.

### 3.10. Membrane Potential Assay

The membrane potential assay was detected by 3,3′-Dipropylthiadicarbocyanine Iodide (DiSC3 (5), MCE, Shanghai, China). The overnight culture *S. aureus* cells were diluted to OD_600_ of 0.3 with a fresh LB medium, then the diluted cells were mixed with DiSC3 (5) at a final concentration of 10 μM and dispensed into black-bottomed 96-well plates. The mixtures were incubated at room temperature until the fluorescence value was stable. Then different concentrations of C_14_ iturin W were supplemented and further incubated for 30 min. The fluorescence of DISC3 (5) was detected at an excitation wavelength of 633 nm and an emission wavelength of 660 nm, using the Infinite M1000 Pro Microplate reader (TECAN, Switzerland). The group untreated with C_14_ iturin W was used as a negative control.

### 3.11. Membrane Permeability Assay

The membrane permeability was determined by propidium iodide (PI, Yuanye, Shanghai, China). The overnight cultures of *S. aureus* cells were diluted to an OD_600_ of 0.3 with PBS and co-cultured with different concentrations of C_14_ iturin W, then PI was added at a final concentration of 20 μg/mL. After incubation in the dark for 15 min at room temperature, the cells were washed 3 times with PBS and resuspended, then the fluorescence intensity was measured using the Infinite M1000 Pro Microplate reader (TECAN, Switzerland) for excitation and emission wavelengths of 535 and 615 nm, respectively. Each group was replicated in triple and the group untreated with C_14_ iturin W was used as a negative control.

### 3.12. Effect of C_14_ Iturin W on the Hemolytic Activity of S. aureus

The hemolysin is one of the important exotoxins of the foodborne pathogen *S. aureus* and can be used as a target to reduce the virulence of *S. aureus* [45]. In order to detect the effect of the C_14_ iturin W on hemolytic activity of *S. aureus*, the *S. aureus* cells were cultured in an LB medium with different concentrations of C_14_ iturin W at 37 °C overnight, and the supernatant was obtained by centrifugation at 9000 rpm for 10 min, then 180 μL of the supernatant of *S. aureus* and 20 μL of sheep erythrocytes (Yuanye, Shanghai, China) were added to the 96-well plate. The 96-well plate was incubated at 37 °C for 1 h, and the OD value was measured at 540 nm using the Infinite M1000 Pro Microplate reader (TECAN, Switzerland). The hemolytic activity assay of C_14_ iturin W itself was also detected, and the groups treated with 180 μL of PBS and 0.1% Triton X-100 were used as negative and positive controls, respectively. Erythrocyte hemolysis rate (%) = (A − A_0_)/(A_1_ − A_0_) × 100%. A is the absorbance of the sample to be tested, A_0_ is the absorbance of the PBS-treated (negative control) group and A_1_ is the absorbance of the 0.1% Triton X-100-treated (positive control) group.

### 3.13. Effect of C_14_ Iturin W on the Production of SEs

In order to detect the effect of C_14_ iturin W on the production of enterotoxins produced by *S. aureus*, the overnight culture of *S. aureus* was diluted into an LB medium to a final concentration of 10^−4^, then different concentrations of C_14_ iturin W were added, and the culture was incubated at 150 rpm at 37 °C for 16–18 h. The cell-free supernatant was obtained by centrifugation at 9000 rpm for 10 min, and the total content of enterotoxins in the supernatant was tested using a *S. aureus* enterotoxins ELISA kit (Zhongxi Biotechnology Co., Ltd., Qingdao, China) according to the manufacturer’s instructions. The OD value was measured at 450 nm using the Infinite M1000 Pro Microplate reader (TECAN, Switzerland). Each sample contained three replicates.

### 3.14. Statistical Analysis

Statistical analysis of each biochemical datum was performed by *t*-test. Each experiment was performed in triplicate; the results of independent experiments were averaged, and the standard error of the mean (mean ± SD) was calculated. “**” indicates that the *p*-value is less than 0.01, “***” indicates that the *p*-value is less than 0.001.

## 4. Conclusions

In summary, our study showed that lipopeptide C_14_ iturin W exhibited strong inhibitory effect on *S. aureus*. In addition, C_14_ iturin W also has the ability to reduce the production of virulence factors of *S. aureus*, such as SEs and hemolysins. Therefore, lipopeptide C_14_ iturin W has a good application potential in the future. The corresponding mechanism was also investigated, and it showed that C_14_ iturin W exerted its antimicrobial activity by down-regulating ROS clearance enzymes, inhibiting the arginine biosynthesis and disturbing the proton motive force, thus leading to changes in membrane permeability and cell death.

## Figures and Tables

**Figure 1 ijms-25-09949-f001:**
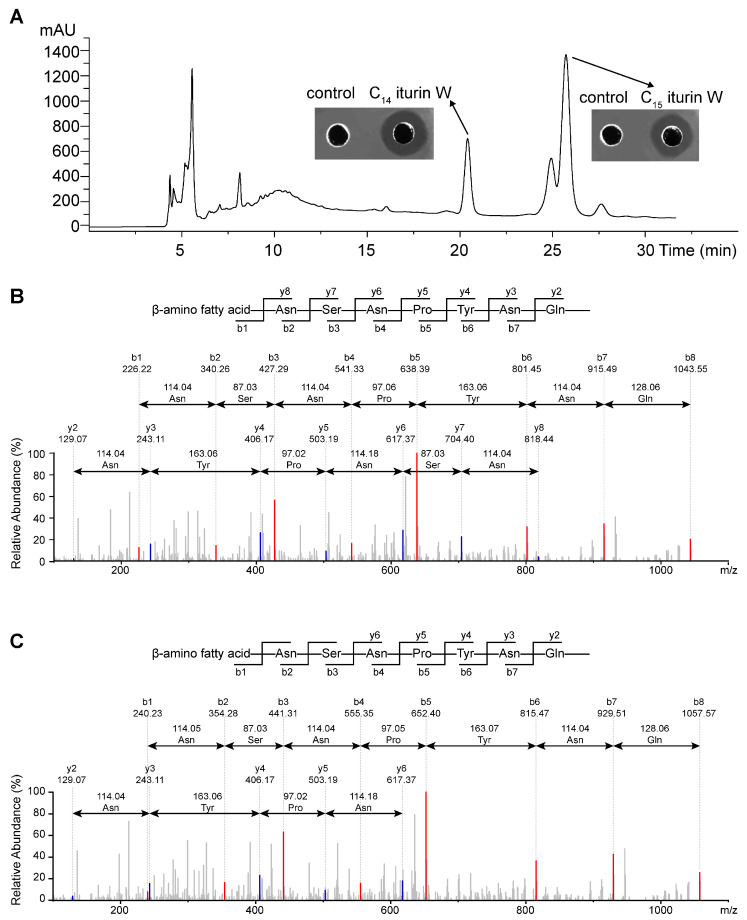
Purification and identification of iturin W by RP-HPLC (**A**) and MS/MS analysis of the antimicrobial agents eluted at 20.420 min (**B**) and 25.725 min (**C**) in (**A**).

**Figure 2 ijms-25-09949-f002:**
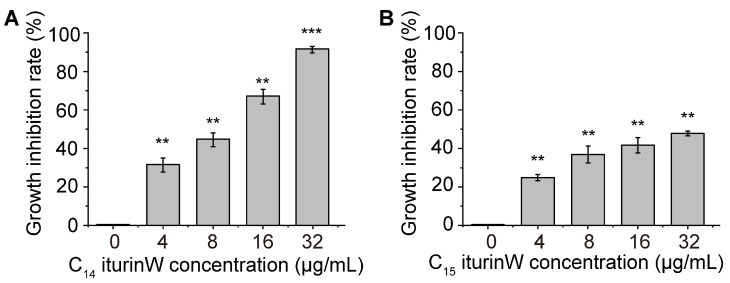
Growth inhibition rate of C_14_ iturin W (**A**) and C_15_ iturin W (**B**) against *S. aureus*. ** *p* < 0.01, *** *p* < 0.001.

**Figure 3 ijms-25-09949-f003:**
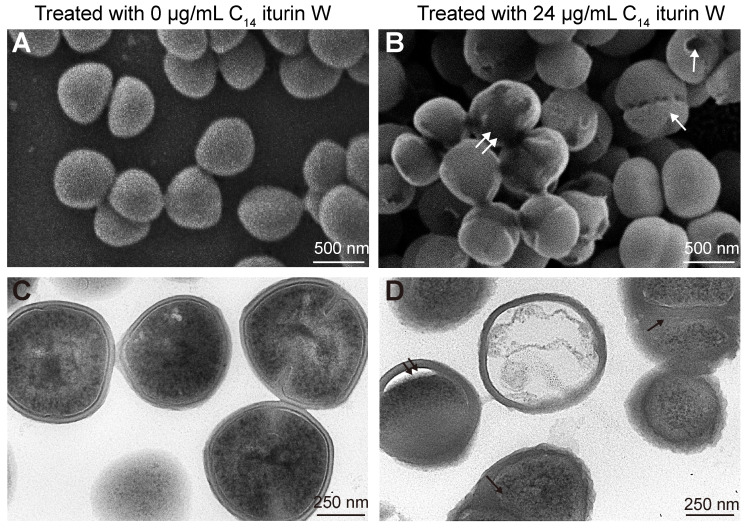
Morphological and ultrastructural changes of MRSA CCARM 3090 cells caused by C_14_ iturin W under SEM (**A**,**B**) and TEM (**C**,**D**) observations. White single arrows indicate the hole on the cell surface, and the white double arrows indicate the aggregated cells. Black single arrows indicate the swollen cell wall, and black double arrows indicate the gap between the cell wall and cell membrane (Magnifications range: SEM 20,000×; TEM 60,000×).

**Figure 4 ijms-25-09949-f004:**
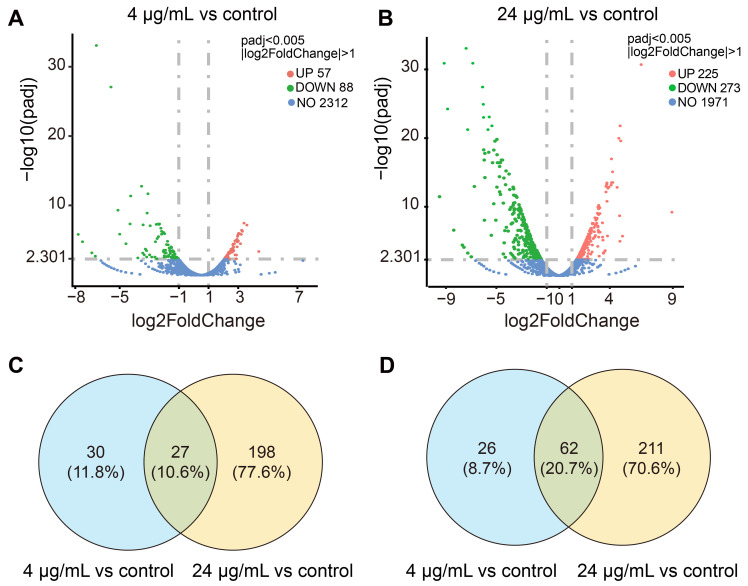
The volcano plot of the DEGs of 4 µg/mL (**A**) and 24 µg/mL (**B**) C_14_ iturin W-treated MRSA CCARM 3090. And Venn diagrams of significant up-regulation (**C**) and down-regulation (**D**) of common and specific DEGs between the two treatment groups.

**Figure 5 ijms-25-09949-f005:**
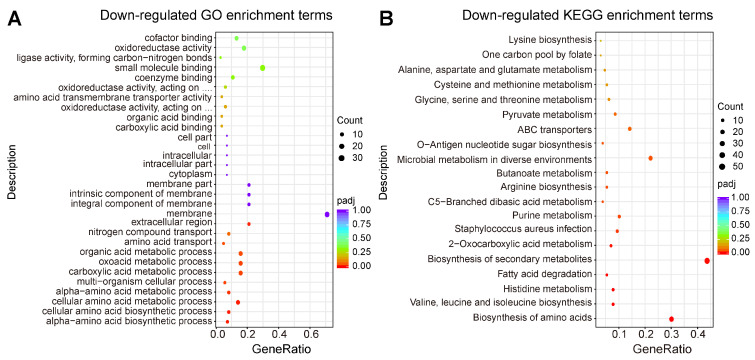
The GO enrichment scatter plot (**A**) and KEGG enrichment scatter plot (**B**) of down-regulated DEGs in MRSA CCARM 3090 cells between the group without C_14_ iturin W treatment and the group treated with 24 µg/mL of C_14_ iturin W.

**Figure 6 ijms-25-09949-f006:**
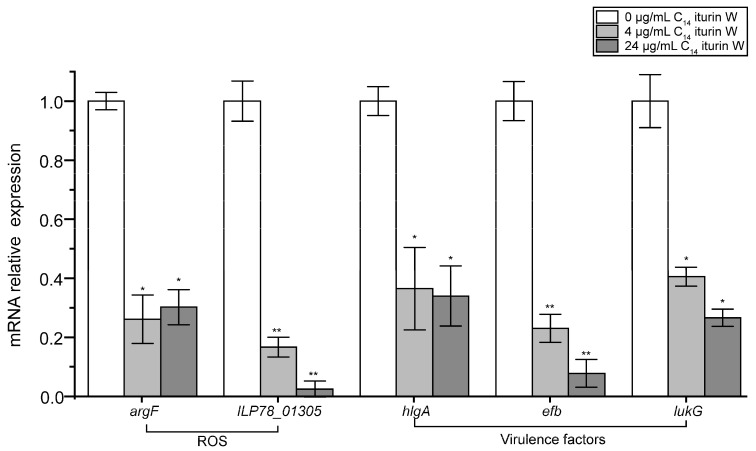
qRT-PCR transcription analysis of the genes related to ROSs and virulence factors. * *p* < 0.05, ** *p* < 0.01.

**Figure 7 ijms-25-09949-f007:**
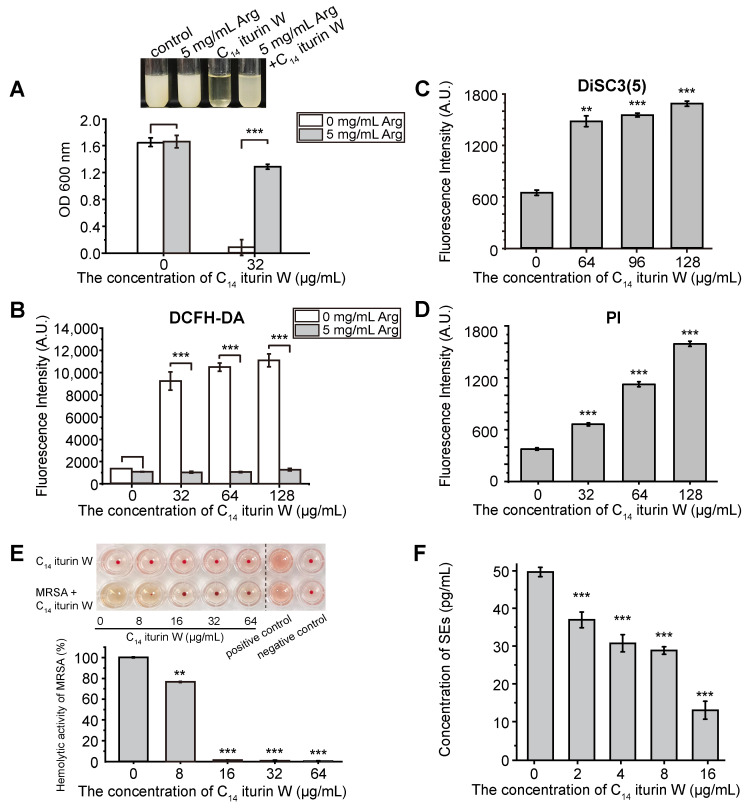
Biochemical functional validation of DEGs. (**A**) The effect of C_14_ iturin W on the growth of MRSA CCARM 3090 cells in the presence of arginine. (**B**) ROS detection caused by different concentrations of C_14_ iturin W with or without arginine supplements. (**C**) Detection of the membrane potential caused by different concentrations of C_14_ iturin W. (**D**) Detection of the membrane permeability of MRSA CCARM 3090 caused by different concentrations of C_14_ iturin W. (**E**) The inhibitory effect of C_14_ iturin W on the hemolytic activity of MRSA CCARM 3090 and (**F**) the production of total SEs. ** *p* < 0.01, *** *p* < 0.001.

**Table 1 ijms-25-09949-t001:** MIC of C_14_ iturin W from Bacillus against selected bacteria, including antibiotic-resistant strains.

Bacterial Strain ^1^	MIC (μg/mL)
MRSA CCARM 3090	64
MRSA QD-1	64
*S. aureus* QD-2	32
*S. aureus* ATCC 25923	32

^1^ CCARM, Korea Seoul Women’s University; QD, the Biological Experiment Teaching Center of Qingdao University (Qingdao, China); ATCC, American Type Culture Collection.

**Table 2 ijms-25-09949-t002:** DEGs related to ROSs and virulence factors in MRSA CCARM 3090 cells after treated with different concentrations of C_14_ iturin W.

Gene	4 μg/mL log_2_FoldChange	24 μg/mL log_2_FoldChange	Gene Description
ROS-related genes			
*ILP78_10355*	−6.115	−7.143	argininosuccinate synthase
*argH*	−7.100	−6.613	argininosuccinate lyase
*argF*	−1.965	−2.346	ornithine carbamoyltransferase
*ILP78_00430*	−2.230	−2.689	superoxide dismutase
*ILP78_01305*	−1.248	−1.693	glutathione peroxidase
proton motive force			
*mnhA2*	−0.776	−1.207	Na^+^/H^+^ antiporter Mnh2 subunit A
*mnhB2*	−0.618	−1.087	Na^+^/H^+^ antiporter Mnh2 subunit B
virulence factor-related genes			
*hlgA*	−1.408	−4.341	bi-component gamma-hemolysin HlgAB subunit A
*hlgB*	−1.872	−4.502	bi-component gamma-hemolysin HlgAB/HlgCB subunit B
*hlgC*	−1.795	−5.407	bi-component gamma-hemolysin HlgCB subunit C
*hyl*	−2.700	−4.238	alpha-hemolysin
*spa*	−2.869	−4.204	staphylococcal protein A
*efb*	−3.676	−5.711	complement convertase inhibitor Efb
*ecb*	−2.695	−4.651	complement convertase inhibitor Ecb
*lukG*	−1.933	−3.232	bi-component leukocidin LukGH subunit G
*lukH*	−0.947	−1.595	bi-component leukocidin LukGH subunit H
*adsA*	−1.439	−1.701	LPXTG-anchored adenosine synthase AdsA

**Table 3 ijms-25-09949-t003:** qRT-PCR primer sequences used in this study.

Primers	Sequence (5′ → 3′)
16S-F	CGCGTGAGTGATGAAGGTCT
16S-R	ATTCCGGATAACGCTTGCCA
*icaA*-F	GTGAAACGATTGAAGATACG
*icaA*-R	GCTTTACCTCTGTTTTCTTG
*icaB*-F	CTTATGGCTTGATGAATGAC
*icaB*-R	GACTGCTTTTTCCTCTAATG
*fnbA*-F	AGATGAACTACCTGAAGAAC
*fnbA*-R	CCATTTTCAGTTCCTAAACC
*fnbB*-F	AGTGGATGAATTACCTGAAG
*fnbB*-R	CCATTTTCAGTTCCTAAACC
*ILP78_12790*-F	GATGGATCAAAGAACTTGAG
*ILP78_12790*-R	CTTTTAGCATGGAAAGGAAG
*ILP78_01305*-F	ATATAAGGATCGTGGGTTTG
*ILP78_01305*-R	CCGTTCACAGATATTTTAGC
*argF*-F	CCTGATGAAGTATGGAAAGA
*argF*-R	CGTATCAGCATTATGGAAAG
*hlgA*-F	GACTATTTCGTCCCAGATAA
*hlgA*-R	CTCGCTTTTATCACCTTTAC
*efb*-F	TAACAATAGCGGCAATAGG
*efb*-R	CTCACTGGTTTCTTTTCTCT
*lukG*-F	AAAGGAACAATAGGTAGTGG
*lukG*-R	ACTTCTCTTGATTCATCCTG

## Data Availability

All transcriptomic data are deposited in the Sequence Read Archive (SRA) under BioProject PRJNA954130 in the National Center for Biotechnology Information (NCBI) database.

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
