# Peer review of "Transcriptomic and Biochemical Analysis of the Antimicrobial Mechanism of Lipopeptide Iturin W against Staphylococcus aureus"

_ijms, 2024, doi:10.3390/ijms25189949_

Round 1

Reviewer 1 Report

Comments and Suggestions for Authors

Dear Authors.

The submitted manuscript presents the evaluation of antistaphylococcal potential of antimicrobial peptide derived from Bacillus sp. wsm-1. To do this several assays were utilized, including transcriptomic profiling, gene expression and impact of tested compund on virulence. factors activity. The presented conclusions correspond with obtained results. Therefore, there are some considerations that need to be reviewed. There are listed below.

1. It would be beneficial to present the inhibition zones associated with screening of strains with inhibitory activity against S. aureus

2. Authors should consider providing the figure presenting the structure of C14 iturin W and C15 iturin W as well as table with sequennces provided, average mass (Da), Monoisotopic mass (Da),  MS analysis (za ;  mz Calc. m/z Found).

3.Growth inhibition rate of C14 iturin W (A) and C15 iturin W (B) against S. aureus is unclear to me. In Material and Methods section authors  refer to works where the MIC values were determined. Here Authors base on the comparison (%) between untreated bacteria and those exposed to the tested compound.

4. Moreover, I do not underestimate the value of obtained results, however the use of only 1 MRSA indicator strains should be justified. Why this strain was used. It should be discussed.

5. Evaluation of antimicrobial activity needs be followed by dilution methods conducted according to CLSI protocols.  Authors refer to 2 works (30,31), however in both of them 2 different ways of antimicrobial activity determination were used. Here, as I presume, the estimation of antimicrobial activity is based on OD600 comparison. 

As the work focuses on general antistaphylococcal activity I would recommend to supply the work on MIC determination (according to CLSI protocol: MH broth, 96well plates etc.) on group of reference (or clinical) strains of S. aureus. 

6. Please provide the source of erythrocytes for hemolysis assays. Were they commercially obtained human RBCs or from donors? This should be reviewed. (If the blood was obtained from donors, the appropriate ethical committee consent number should be given)

7. There are more than 20 types of enterotoxins. What was the specificity of ELISA assay? 

8. What software was used for statistical analyses or generation of figures/scatter/volcano plots and venn diagrams? This should be provided. 

9. To sum up. Authors conclude that  C14 iturin W exhibited strong inhibitory activity on the  growth of S. aureus". In my opinion such conclusion could be drawn only when the compound was tested on wide-range of clinical and reference strains. I highly recommend providing such data. Another issue are some grammar errors that needs checking

Comments on the Quality of English Language

Line 95: "the centration of 32 μg/mL, the inhibition rate..."

Line 428: "..our study showed that lipopeptide C14 iturin W exhibited strong inhib-427 itory on the cell growth of S. aureu."

Reviewer 2 Report

Comments and Suggestions for Authors

Introduction Introduction is too short. S. aureus is one of the most important foodborne pathogens, … and is a major reason of food poisoning worldwide - Top five pathogens contributing to foodborne illnesses are: Norovirus, Salmonella, nontyphoidal, Clostridium perfringens, Campylobacter spp. and Staphylococcus aureus, only 3% of all cases. In whole methods it is not clear which strain, MRSA (CCARM 3090) or indicator strain S. aureus was used. How many repetitions were performed for all analyses? It is not clear which bacteria produce iturin, Bacillus sp. – please add full name of species Figures 1, 5 are not legible. 2. Results and Discussion must be separated. 3.1. Screening of strains with inhibitory activity against S. aureus In order to screen bacteria that can inhibit S. aureus, the inhibitory effect of 200 marine strains isolated and purified in our laboratory were detected according to previous method – all strains name should added as supplementary materials, moreover it is self-citation (reference 30) Lack of S. aureus origin – MRSA (CCARM 3090) and indicator strain S. aureus 100 μL of fermentation supernatant of the isolated strain was added into the pre-punctured wells in LB plate – all 200 marine strains were tested? Moreover details about fermentation supernatant obtaining should be added. 28 ℃ for 12 h – for S. aureus incubation is usually 35±1°C, 18±2h. Please explain incubation condition in lower temperature and shorter time. Moreover why indicator strain S. aureus was incubated in 35°C? Moreover please add incubation time. Line 282 - …strain with clear inhibitory zone was considered to have inhibitory activity against S. aureus – please add details about inhibitory zone evaluation (measured in mm?) 3.2. Purification of the antimicrobial agent against S. aureus The purification of the antimicrobial agent against S. aureus from the screened strain was carried out according to previous method – please add details about antimicrobial agent - moreover it is self-citation (reference 11) Line 289 - screened strain – please add name of strains Line 292 - precipitate was collected by centrifugation – please add condition 3.3. Identification of the purified antimicrobial agent To identify the purified antimicrobial agent, … as we described previously, it is self-citation (reference 11) 3.4. Antimicrobial activity assay Line 312 - S. aureus was diluted to 104-5 CFU/mL – why Authors used 104 or 105 CFU/mL? Moreover which strain was used? MRSA (CCARM 3090) or indicator strain S. aureus? 3.5. Scanning electron microscopic (SEM) and transmission electron microscopic (TEM) observation Line 322 - The culture of S. aureus was diluted – please add dilution and next name of antimicrobial agent, incubation and centrifugation condition, and microscope name 3.6. Transcriptomic analysis Please add name of antimicrobial agent, incubation condition 3.7. Quantitative real-time reverse transcription PCR (qRT-PCR) analysis Please add name of antimicrobial agent. Moreover please add program to designed primers listed in Table 2. Moreover add Real-Time PCR System name and producer 3.8. Detection of the correlation between arginine and the antimicrobial agent Why arginine was used? Moreover why Authors used 104 or 105 CFU/mL? Please add name of antimicrobial agent, incubation condition. Line 366 - cell growth was detected according to above methods. Which method? 3.9. Determination of ROS Please add spectrophotometer name and producer 3.10. Membrane potential assay Please add fluorescence reader name and producer 3.11. Membrane permeability assay Please add microplate reader name and producer 3.12. Effect of antimicrobial agent on the hemolytic activity of S. aureus Please add name of antimicrobial agent and spectrophotometer name and producer Line 410 - A0 is the absorbance of the negative control group, A1 is the absorbance of the positive control group. Please add detail A0 and A1 control group 3.13. Effect of antimicrobial agent on the production of SEs Please add name of antimicrobial agent and spectrophotometer name and producer 4. Conclusions Therefore, lipopeptide C14 iturin W has a good application potential in food safety in the future - based on what study? This is an over interpretation on the basis of few in vitro analysis. Lack of reagent and media producer Almost half of the cited articles are outdated. 16 from 31 older than 5 years.

Reviewer 3 Report

Comments and Suggestions for Authors

Minor revisions:

lines 15-19: Explain the primary antimicrobial mechanism of lipopeptide Iturin W against Staphylococcus aureus as discussed in the study?

lines 16-20 and 184-206: How does the study describe the impact of C14 Iturin W on the expression of genes associated with virulence factors in Staphylococcus aureus?

lines 105-118:What were the key findings from the morphological and ultrastructural changes observed in Staphylococcus aureus cells treated with C14 Iturin W?

lines 125-149: Explain the results of the transcriptomic profiling performed on Staphylococcus aureus after treatment with C14 Iturin W.

lines 159-183:What role do reactive oxygen species (ROS) and proton motive force play in the antimicrobial activity of C14 Iturin W according to the study?

lines 209-215: How did the study validate the transcriptomic data through qRT-PCR, and what were the main findings?

lines 258-267 and 429-431: Discuss the potential applications of C14 Iturin W in food safety as mentioned in the study.

 lines 16-20 and 184-206: How does the study describe the impact of C14 Iturin W on the expression of genes associated with virulence factors in Staphylococcus aureus?

lines 105-118: What were the key findings from the morphological and ultrastructural changes observed in Staphylococcus aureus cells treated with C14 Iturin W?

lines 125-149: Explain the results of the transcriptomic profiling performed on Staphylococcus aureus after treatment with C14 Iturin W.

lines 159-183: What role do reactive oxygen species (ROS) and proton motive force play in the antimicrobial activity of C14 Iturin W according to the study?
